# Assessing Trifecta and Pentafecta Success Rates between Robot-Assisted vs. Open Radical Cystectomy: A Propensity Score-Matched Analysis

**DOI:** 10.3390/cancers16071270

**Published:** 2024-03-25

**Authors:** Filippo Gavi, Nazario Foschi, Daniele Fettucciari, Pierluigi Russo, Diana Giannarelli, Mauro Ragonese, Carlo Gandi, Giovanni Balocchi, Alessandra Francocci, Francesco Pio Bizzarri, Filippo Marino, Giovanni Battista Filomena, Giuseppe Palermo, Angelo Totaro, Marco Racioppi, Riccardo Bientinesi, Emilio Sacco

**Affiliations:** 1Postgraduate School of Urology, Catholic University Medical School, Largo Francesco 6 Vito 1, 00168 Rome, Italy; filippo.gavi@guest.policlinicogemelli.it (F.G.); francescopio.bizzarri01@icatt.it (F.P.B.); filippo.marino01@icatt.it (F.M.); marco.racioppi@policlinicogemelli.it (M.R.);; 2Department of Urology, Fondazione Policlinico Universitario Agostino Gemelli IRCCS, Largo Francesco 8 Vito 1, 00168 Rome, Italygiuseppe.palermo@policlinicogemelli.it (G.P.);; 3Facility of Epidemiology and Biostatistics, Fondazione Policlinico Universitario Agostino Gemelli IRCCS, Largo Francesco 8 Vito 1, 00168 Rome, Italy; 4Urology Department, Isola Tiberina Gemelli Isola Hospital, Catholic University Medical School, 00168 Rome, Italy

**Keywords:** bladder cancer, open radical cystectomy, propensity score matched, robot-assisted radical cystectomy, robotic surgery

## Abstract

**Simple Summary:**

The use of robot-assisted radical cystectomy (RARC) is increasing, although its definitive superiority over open radical cystectomy (ORC) has not been proven. Randomized controlled trials (RCTs) have demonstrated RARC’s superiority regarding perioperative and functional outcomes but not in terms of oncologic and survival outcomes. Composite outcomes, such as the trifecta and pentafecta, have been proposed to evaluate the quality of the surgery. The aim of our retrospective study was to assess the superiority of RARC over ORC using the PROMETRIC group’s trifecta and pentafecta criteria in a propensity score-matched analysis to reduce biases. No differences were found in the success rates of trifecta and pentafecta. The overall survival was comparable between the two cohorts. We confirmed the superiority of RARC in significantly reducing the estimated blood loss and perioperative transfusion rates.

**Abstract:**

Background: This study aimed to evaluate the surgical and oncological outcomes of robot-assisted radical cystectomy (RARC) versus open radical cystectomy (ORC) using trifecta and pentafecta parameters. Methods: The clinical data of 41 patients who underwent RARC between 2018 and 2022 were prospectively collected and retrospectively compared to those of 330 patients undergoing ORC using 1:1 propensity score matching. Trifecta was defined as simultaneous negative surgical margins (SMs), a lymph node (LN) yield ≥ 16, and the absence of major complications (Clavien–Dindo grade III–V) within 90 days postoperatively. Pentafecta additionally included a 12-month recurrence-free rate and a time between the transurethral resection of a bladder tumor (TURBT) and radical cystectomy (RC) ≤ 3 months. The continuous variables were compared using the Mann–Whitney U test, and the categorical variables were analyzed using the chi-squared test. Results: No statistically significant differences in trifecta and pentafecta success rates were observed between the RARC and ORC cohorts after propensity score matching. However, the RARC group exhibited significantly reduced blood loss (RARC: 317 mL vs. ORC: 525 mL, *p* = 0.01). Conclusions: RARC offers distinct advantages over ORC in terms of reduced blood loss, while trifecta and pentafecta success rates do not differ significantly between the two surgical approaches.

## 1. Introduction

Urothelial bladder cancer is the most frequent type of bladder cancer (BCa). BCa is frequently detected in older people and is closely related to smoking, but also to environmental factors and toxics [1]. BCa is the fifth most common cancer in Europe [2]. BCa is classified as non-muscle-invasive bladder cancer (NMIBC) and muscle-invasive bladder cancer (MIBC). The standard of care for treating high-risk BCa is radical cystectomy (RC), with regional pelvic lymphadenectomy (PLND) preceded by neoadjuvant chemotherapy [3,4]. MIBC, even with optimal care, still presents with high morbidity and mortality [5]. There are major surgical risks associated with conducting extensive pelvic surgery and reconstructing the urinary system in an aged, comorbid population. A wide range of surgical treatments have made extensive use of minimally invasive surgical techniques. Using minimally invasive methods more frequently is one way to reduce surgical morbidity and accelerate recovery [6]. To reduce surgical morbidity, minimally invasive procedures, like robotic-assisted radical cystectomy (RARC), have been developed. Compared to open surgery, robotic-assisted surgery in urology has been proven to reduce blood loss, the requirement for transfusion, and the duration of the stay [7,8,9]. Although these advantages are commonly accepted, no studies have shown the significant impact of RARC compared to ORC regarding oncologic and long-term survival. The Razor study, which is a randomized phase 3 non-inferiority trial, did not find differences between RARC and ORC in 2-year progression free survival [10]. Concerns regarding RARC costs have also been reported [11]. The scientific community needs to finally establish the function of the robotic technique for the surgical treatment of bladder cancer because high-level evidence in favor of robotic RC remains scarce [12]. Several criteria have been proposed to assess RC quality in the past, evaluating oncological and perioperative outcomes [13]. Trifecta and pentafecta criteria are well established for partial nephrectomy (surgical margins status, functional preservation, and complications) [14] and radical prostatectomy (urinary continence, potency, cancer control, surgical margin status, and postoperative complications) [15]. Aziz et al. [16] proposed trifecta and pentafecta criteria composed of oncological and functional outcomes. In this study, we sought to determine whether RARC had a definite advantage over ORC using the trifecta and pentafecta criteria.

## 2. Materials and Methods

### 2.1. Patients

This study was approved by the institutional review board of the Policlinico Agostino Gemelli (N0022214/22). The data of patients treated with ORC or RARC at the Policlinico Agostino Gemelli for BCa between 2017 and 2022 were prospectively collected from the institution’s cystectomy database. The inclusion criteria were age ≥ 18 years and a diagnostic TURBt for BCa with a T stage (cT) 2-4, cN0, cM0 or recurrent BCG failure high-grade cancer. Written informed consent was obtained from all patients. Postoperatively, all patients were followed at 3, 6, and 12 months with routine clinical history and physical exams, diagnostic imaging of the chest/abdomen/pelvis, and complete blood work. The patients’ baseline characteristics included age, gender, body mass index (BMI), smoking behavior, Charlson Comorbidity Index (CCI), the administration of neoadjuvant chemotherapy (NAD), and treatment-free time between TURBt and RC. The perioperative data included urinary diversion type (ileal conduit, orthotopic neobladder, and ureterocutaneostomy), the total operating time (in the RARC group, the time taken for urinary diversion was not evaluated independently, even if it was conducted using the open surgical technique), estimated blood loss (EBL), perioperative transfusion rates (PBTs), complication rate at 90 days classified according to Clavien–Dindo classification [17], length of hospital stay (LOS), hospital readmission rate at 90 days, and overall survival at 12 months (OVS). Oncological outcomes included histology, pathologic stage (TNM classification), soft tissue surgical margins (STSMs), lymph node positivity. and recurrence at 12 months confirmed by radiologic studies during the follow-up.

### 2.2. Surgical Intervention

ORC was performed by one of three surgeons experienced in major oncological surgery. RARC was performed by one of two surgeons experienced in robotic surgery. The extent of the PLND was planned preoperatively based on the international oncological guidelines, the severity of the disease, and the presence of vascular disease. Based on intraoperative clinical results (vascular disease, fibrosis, and adenopathy), the degree of PLND could be changed. The decision to perform orthotopic neobladder, ileal conduit, or ureterocutaneostomy was made by evaluating the patients’ preferences and clinical factors. The exclusion criteria for orthotopic neobladder were urethral neoplasm, extensive tumor burden, cognitive impairment, and inability to perform intermittent catheterization.

### 2.3. Outcomes

The primary endpoint of this study was to assess differences in trifecta and pentafecta success rate between RARC and ORC, which were constructed based on the following criteria by Aziz et al. [16]:(1)Negative soft tissue surgical margins.(2)Lymphadenectomy of ≥16 LNs.(3)Absence of major complications at 90 days.(4)Treatment-free time between TURBt and RC shorter than 3 months.(5)Absence of local recurrence within 12 months after RC.

These five outcomes formed the pentafecta, and the top three outcomes formed the trifecta.

Secondary endpoints were the assessment of significant differences in the estimated blood loss (EBL), perioperative units of blood transfused rates (PBTs), length of hospital stay (LOS), 90-day hospital readmissions, and overall survival (OVS) at 12 months.

### 2.4. Statistical Analysis

The continuous variables were reported as the median and interquartile range (IQR) or mean and standard deviation (sd). The categorical variables were reported as frequencies and proportions. The comparison between the RARC and ORC cohorts was assessed using a chi-squared test for the categorical variables. The continuous variables were compared with a Mann–Whitney U test before PS matching and with a Wilcoxon rank-sum test after PS matching. The Kaplan–Meier method and the log-rank test were applied to compare OVS at 12 months. The statistical analyses were performed using Stata version 18 (Stata Corp., College Station, TX, USA). Statistical significance was set at *p* < 0.05.

#### Propensity Score Matching

To reduce biases in our dataset, we matched 41 patients treated with RARC with 41 patients treated with ORC (1:1 ratio) from a cohort of 330 patients. PS was constructed with a multivariable logistic regression model considering the following variables: age, neoadjuvant chemotherapy (NAD), CCI, and pathologic stage. The matching was calculated using the nearest-neighbor matching algorithm (caliper width 0.25 of the standard deviation of the logit score) with a 1:1 ratio with non-replacement [18].

## 3. Results

Between January 2017 and December 2022, 409 patients were treated with RC for BC; of these, 38 patients were excluded due to lack of follow-up data. In total, 371 patients were included in the study, 41 patients were treated with RARC, and 330 were treated with ORC. The mean operating times were 447 min and 356 min in the RARC and ORC groups, respectively. The median follow-up times were 10 (IQR: 9–12) months after RC. No differences were found in the pathologic stage between the two groups, with 61% (25/41) and 51% (169/330) being T2 in the RARC and ORC groups, respectively. No significant differences in histology were reported between the two groups. A statistically significant difference was observed in the UD type between the groups regarding orthotopic neobladder (RARC: 44% vs. ORC: 17%, *p* < 0.01) and ileal conduit (RARC: 56% vs. ORC: 75%, *p* < 0.01). Similarly, minor (Clavien–Dindo ≤ I–II) complication rates were higher in ORC (RARC: 51% vs. ORC: 86%, *p* < 0.01) (Table 1).

The primary endpoint was to assess differences in trifecta and pentafecta success rates between the ORC and RARC groups. The trifecta success rates were 63% (26/41) and 57% (188/330) for patients treated with RARC and ORC, respectively (*p* = 0.43). The pentafecta success rates were 32% (12/41) and 39% (128/330) of patients treated with RARC and ORC, respectively (*p* = 0.38). No statistically significant differences were found after PS matching in the trifecta and pentafecta success rates. The analyses for each variable of the trifecta and pentafecta before and after PS matching are reported in (Table 2). Before PS matching, there was a statistically significant difference in the treatment-free time ≤ 3 months after TURBT (RARC: 61% vs. ORC: 71%, *p* = 0.01) and in the absence of local recurrence within 12 months after RC (RARC: 85% vs. ORC: 71%, *p* = 0.01). After PS matching, only a statistically significant difference in the treatment-free time ≤ 3 months after TURBT was observed (*p* = 0.02). There were no statistically significant differences pre- and post-PS matching in negative STSMs (RARC: 98% vs. ORC: 96% *p* = 0.43), ≥16 lymph node count (RARC: 85% vs. ORC: 80%, *p* = 0.43), and the absence of major (Clavien–Dindo ≥ III) complication rates (RARC: 15% vs. ORC: 17%, *p* = 0.36) (Table 2).

Pre- and post-PS matching EBLs (RARC: 317 mL vs. ORC 622 mL, *p* = 0.01) and perioperative transfusion rates (RARC: 21% vs. ORC 38%, *p* = 0.01) were significantly lower, while no differences in terms of LOS (RARC: 16 vs. ORC: 15, *p* = 0.60) and 90-day hospital readmission rates (RARC: 20% vs. ORC: 24%, *p* = 0.88) were observed (Table 3).

Upon Kaplan–Meier analysis, no significant differences in OVS between the two cohorts were observed before (log rank = 0.32) or after PS matching (log rank = 0.26) (Figure 1).

## 4. Discussion

Radical cystectomy remains the benchmark treatment for muscle-invasive bladder cancer. ORC has been refined over decades; therefore, the majority of urologists feel confident with this technique. However, RARC is becoming increasingly prevalent and accessible. RARC has been demonstrated to be technically feasible, with several RCTs indicating its potential [19,20,21,22,23,24,25]. Recent meta-analyses reported advantages in LOS, EBLs, and PBTs with RARC [26,27]. Of interest is the association between PBTs and worse oncological outcomes (higher recurrence rates and mortality), which is an important factor to take into account when choosing the optimal surgical approach [28]. ORC has shown a longer LOS and worse early health-related QoL outcomes compared to RARC at 5, 12, and 26 weeks [29]. No significant differences in terms of operative times between RARC with intracorporeal reconstruction (iRARC) and ORC were observed [27]. However, no clear advantages of reducing major complication rates and improving OVS and QoL were found in RARC, although the RCTs evaluating these outcomes did not assess survival outcomes [27]. Therefore, in such a complex scenario, reporting surgical outcomes correctly and precisely is pivotal to adequately compare the two approaches. To help to standardize outcomes in robotic surgery, Salomon et al. proposed the trifecta for radical prostatectomy in 2003 [30]. In 2015, Aziz et al. [16] (PROMETRICS group) introduced the concepts of trifecta and pentafecta to assess the quality and oncological efficacy of radical cystectomy [16,31]. In 2019, Cacciamani et al. [31] expanded upon this by proposing an RC-pentafecta, which includes the absence of ureteral diversion-related surgical complications. We tested the trifecta and pentafecta from the PROMETRICS group on both the RARC and ORC cohorts. Our study revealed no differences in trifecta (RARC: 67% vs. ORC: 57%, *p* = 0.43) and pentafecta (RARC: 32% vs. ORC: 39%, *p* = 0.38) success rates between the RARC and ORC cohorts. The trifecta and pentafecta helped us in assessing RC quality, but functional outcomes and QoL were not considered, which is something to consider in the future. 

The rates of negative STSMs were similar between the two groups (RARC: 98% vs. ORC: 96%, *p* = 0.62). Negative STSMs are a critical outcome in oncologic surgery, being associated with metastatic progression, overall survival, and cancer-specific mortality [32]. In the literature, a positive STSM rate < 10% and a lymph node count ≥ 16 is considered as a quality outcome [33]. No significant differences in the lymph node count were observed (RARC: 85% vs. ORC: 80%, *p* = 0.43). After PSM analysis, no differences were found between the two cohorts. These rates are in line with previous studies on RARC vs. ORC [20,21,24,26]. 

No statistically significant differences were observed regarding the absence of major complications at 90 days, while the interval between the transurethral resection of bladder tumor and radical cystectomy was notably briefer among patients in the ORC group (RARC: 61% vs. ORC: 78%, *p* = 0.01). This discrepancy likely derived from the limited availability of robotic surgical slots at our institution. Additionally, the COVID-19 pandemic may have exacerbated this issue by reducing the number of robotic procedures performed, thereby impacting surgical waiting lists adversely [34,35].

After PS matching, RARC still demonstrated superior outcomes in terms of intraoperative EBLs and decreased rates of PBTs, yet no further significant disparities were identified. Notably, a reduction of over 50% in blood loss has been documented, consistent with previous studies [22,24,25,29].

Because RC is associated with significant morbidity [36], the use of minimal invasive approaches is promising to reduce perioperative complications and increase functional outcomes [37]. Recovery protocols have been proposed to reduce peri- and intraoperative complications through multidisciplinary efforts [38]. Enhanced recovery after surgery (ERAS) protocols aim to optimize recovery reducing perioperative complications, time to first bowel movement, and the length of hospital stay [39]. However, no significant differences were found in LOS (RARC: 16 days vs. ORC: 15 days, *p* = 0.53), which is in line with the results from the BORARC trial [21].

OVS rates at 12-month follow-up for RARC and ORC were 93% and 84% (*p* = 0.29), respectively, which is in line with the current literature. We observed no differences in OVS after PS matching and Kaplan–Meier survival analysis (log rank = 0.26). However, this study did not assess survival outcomes; therefore, a larger sample and longer follow-up times are needed. Interestingly, Brassetti et al. [37] in a retrospective study found a significant difference in OVS in patients achieving trifecta at 12 months, defined as urinary continence, recurrence-free status (RFS), and the absence of RARC-/ICUD-related severe complications (SCs) after iRARC, suggesting that trifecta could be used as a tool to decide which patients could benefit from a stricter follow-up. In general, OVS in the first years is commonly associated with bladder cancer, but with a longer follow-up time, deaths are associated with patients’ comorbidities or other diseases not related to bladder cancer [3].

Readmission rates were comparable between the two cohorts (RARC: 20% vs. ORC: 21%, *p* = 0.90), and similar outcomes were reported in previous studies [29]. The main causes of readmission in the first 90 days were infection, sepsis, and surgical wound issues. At 90 days, readmission rates were around 26%, of which 60% occurred in the first 30 days. Unfortunately, no clear benefits have been proven on the role of RARC in reducing readmission rates [27,40].

In our cohorts, UD assessment were performed extracorporeally, which could have reduced the benefits of a totally robotic procedure [37]. Nonetheless, a recent RCT between ORC and RARC with a totally intracorporeal urinary reconstruction only showed similar oncological outcomes, but a significant impact on perioperative outcomes [41].

This study is subject to several limitations. Firstly, its retrospective nature and single-center design conducted by experienced surgeons in both approaches may limit the generalizability of the surgical outcomes. Additionally, the RARC cohort’s small size, although mitigated to some extent by propensity score matching, underscores the need for a larger sample size to bolster the validity of the findings. Due to the limited number of patients in the RARC cohort, we were unable to conduct an optimal stratified analysis based on the urinary diversion type. Consequently, the length of hospital stays may have been influenced by the higher frequency of neobladder reconstruction in patients undergoing RARC, who typically require hospitalization for at least 10–12 days. Moreover, even if our surgeons were experienced in robotic surgery, they have performed more ORC than RARC in their career, and the learning curve for RARC is not clear; therefore, complication rates could further decrease in the future using the robotic approach.

## 5. Conclusions

Patients undergoing RARC exhibited significantly lower intraoperative blood loss and perioperative transfusion rates, highlighting RARC’s notable benefits over ORC in minimizing blood loss. Nonetheless, trifecta and pentafecta success rates did not vary between the two groups. A longer-term follow-up and a larger sample size are imperative to comprehensively evaluate differences in oncologic and functional outcomes between the two surgical approaches.

## Figures and Tables

**Figure 1 cancers-16-01270-f001:**
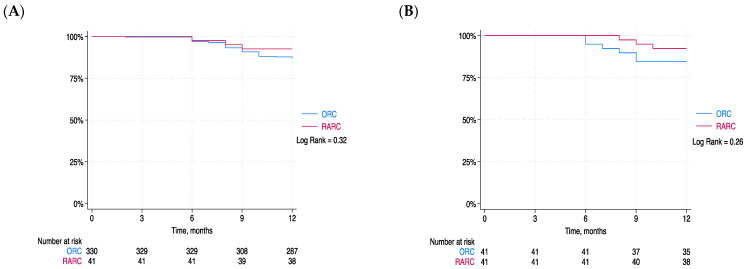
Blue lines represent patients who underwent open radical cystectomy (ORC) and red lines those who underwent robot-assisted radical cystectomy (RARC). (**A**) Kaplan–Meier curves comparing overall survival (OS) before PS matching (log rank = 0.32). (**B**) Kaplan–Meier curves comparing overall survival (OS) after PS matching (log rank = 0.26).

**Table 1 cancers-16-01270-t001:** Patients’ characteristics.

Variables	RARC (*n* = 41)	ORC (*n* = 330)	*p*-Value
Male sex, *n* (%)	33 (80)	262 (80)	0.89
Median BMI, kg/m^2^ (IQR)	25 (18–30)	25 (17–40)	0.96
Median age, yr (IQR)	62 (45–76)	70 (43–93)	0.001
Smokers, *n* (%)	32 (78)	259 (79)	0.78
Charlson comorbidity index, *n* (%)			0.35
0–1	2 (5)	25 (7)	
2–3	16 (39)	125 (38)	
≥4	23 (56)	180 (55)	
Neadjuvant chemotherapy, *n* (%)	22 (53)	123 (38)	0.12
Urinary diversion type, *n* (%)			0.01
Ileal conduit, *n* (%)	23 (56)	246 (75)	
Orthotopic neobladder, *n* (%)	18 (44)	57 (17)	
Ureterocutaneostomy, *n* (%)	0	26 (8)	
Total operating time, min			0.001
Median (IQR)	442 (360–522)	351 (304–404)	
Mean *±* sd	447 *±* 90	356 *±* 80	
Pathologic stage, *n* (%)			0.44
T0	0	12 (4)	
Tis	4 (10)	36 (11)	
T1	6 (15)	31 (9)	
T2	25 (61)	169 (51)	
T3	4 (10)	54 (17)	
T4	2 (5)	28 (8)	
Histology, *n* (%)			0.44
Urothelial cell carcinoma	34 (84)	289 (88)	
Squamous cell carcinoma	2 (5)	17 (5)	
Adenocarcinoma	0	2 (1)	
Other	5 (14)	22 (7)	
Lymph node-positive patients, *n* (%)	7 (17)	40 (16)	0.53
No. 90-day complications, *n* (%):			0.01
Clavien–Dindo I–II	21 (51)	285 (86)	
Clavien–Dindo III–IV	6 (15)	45 (14)	

BMI = body mass index; IQR: interquartile range; ORC = open radical cystectomy; RARC = robot-assisted radical cystectomy.

**Table 2 cancers-16-01270-t002:** RARC vs. ORC pentafecta and trifecta success rates.

	Before PS Matching			After 1:1 PS Matching	
	RARC (*n* = 41)	ORC (*n* = 330)	*p*-Value	ORC (*n* = 41)	*p*-Value
Trifecta, *n* (%)	26 (63)	188 (57)	0.43	27 (65)	0.46
Pentafecta, *n* (%)	13 (32)	128 (39)	0.38	13 (32)	0.86
Negative STSMs, *n* (%)	40 (98)	317 (96)	0.62	37 (96)	0.88
Lymph node count ≥ 16, *n* (%)	35 (85)	265 (80)	0.43	33 (82)	0.32
Absence of Clavien–Dindo grade ≥ III complications at 90 days, *n* (%)	35 (85)	284 (86)	0.36	36 (88)	0.22
≤3 months between TURBT and RC, *n* (%)	25 (61)	258 (78)	0.01	30 (72)	0.02
Absence of local recurrence within 12 months after RC, *n* (%)	35 (85)	237 (71)	0.01	34 (83)	0.55

STSMs = soft tissue surgical margins; ORC = open radical cystectomy; RARC = robot-assisted radical cystectomy.

**Table 3 cancers-16-01270-t003:** RARC vs. ORC secondary endpoints.

	Before PS Matching			After 1:1 PS Matching	
	RARC (*n* = 41)	ORC (*n* = 330)	*p* Value	ORC (*n* = 41)	*p* Value
Lenght of hospital stay, mean ± sd	16 *±* 11	15 *±* 12	0.53	15 *±* 11	0.60
Mean estimated blood loss, mL ± sd	317 *±* 26	622 *±* 22	0.01	525 *±* 65	0.01
Perioperative transfusion, *n* (%)	9 (21)	125 (38)	0.01	18 (42)	0.01
90-day hospital readmissions, *n* (%)	8 (20)	70 (21)	0.90	10 (24)	0.88

ORC = open radical cystectomy; RARC = robot-assisted radical cystectomy.

## Data Availability

All data used in this analysis were sourced from an anonymized database. The code for the analyses can be provided upon request.

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
