# Peer review of "Assessing Trifecta and Pentafecta Success Rates between Robot-Assisted vs. Open Radical Cystectomy: A Propensity Score-Matched Analysis"

_cancers, 2024, doi:10.3390/cancers16071270_

Round 1

Reviewer 1 Report

Comments and Suggestions for Authors

Using a single-center database, this paper aims to compare the oncologic outcomes of robotic-assisted radical cystectomy (RARC) versus open radical cystectomy (ORC).

The primary endpoints are achievement of trifecta and pentafecta, composite endpoints widely used in urologic oncologic surgery. Secondary endpoints are blood loss during surgery, units of blood transfused, length of hospital stay, hospital readmission at 90 days, and overall survival at 12 months.

The authors performed propensity score matching (PSM) to avoid selection bias and to make the RARC and ORC cohorts comparable in terms of preoperative characteristics and risk factors.

The study design is simple, as are the statistical analyses, except perhaps for PSM, which is more subtle in its implementation.

Unfortunately, the work in the "Materials and Methods" and "Results" sections is haphazard and repetitive, forcing the reader to jump from paragraph to paragraph to maintain the linearity of the information. Errors also indicate haste and gradually make the reader lose confidence in the study.

The following are the main problems

Introduction

1.      There is an unnecessary repetition of trifecta in the definition of the criteria that make up pentafecta (lines 67-70.) This redundancy also recurs in the outcomes section (lines 111-113).

2.      RFS acronym (line 70) is not explained.

Materials and Methods

3.      The study design that makes the reader understand the importance of the work is missing.

Patients

4.      The list of database variables (lines 81 to 92) is largely redundant, given their presence in the results tables. A description of how the data are collected and verified in their reliability would be more informative at this point.

Surgical intervention

5.      The phrase "and if hysterectomy and bilateral salpingo-oophorectomy were present, they were removed in women" (lines 96-97) is obscure to me. It is necessary to clarify this point.

Outcomes

6.      How do the indicators "no major complication at 90 days" (major outcome) and "readmission at 90 days" (secondary endpoint) differ? Was readmission (unplanned?) within 90 days considered a major complication?

Propensity score matching

7.      PSM should be included in the statistical analysis paragraph. 

8.      Why were sex, smoking and BMI excluded from PSM calculation?

Statistical analysis

9.      How come continuous variables are reported as mean and interquartile range? (line 131).

10.  I am unclear about the statistical approach to assessing differences in continuous variables. Have Wilkoxon rank-sum test and Mann-Whitney U test (line 135) been applied under different conditions?

Results

11.  Table 1 does not show the characteristics of patients before and after PSM (rows 127-128). The columns in Table 1 are for 41 patients in the RARC group and 330 patients in the ORC group.

12.  Table 1 contains several errors, please correct.

13.  In addition to the primary outcomes for the RARC and ORC cohorts after PSM, Table 2 also shows the primary outcomes of the entire ORC cohort (330 patients). Is there a reason for this? This table is also described in detail in section 3.1 of the text (lines 160-176), without any reasons or differences appearing to justify this choice.

14.  Figures 1 and 2 show the trifecta and pentafecta endpoints already shown in Table 2. In addition, these figures can be matched into one.

15.  Table 3 contains errors.

Author Response

Dear Reviewer,

Thank you for your review. We have addressed each comment, on a point-by-point basis. Accordingly, we have included the revised manuscript for your review.

Warm regards,

Filippo Gavi

Reviewer 2 Report

Comments and Suggestions for Authors

This is a comparative paper between open and robotic-assisted surgery for total cystectomy.

As you mentioned, the time from TURBT to surgery may be difficult to compare with previous open surgeries because of the possible influence of COVID19.

There is no significant difference, but there may be a difference because there are more NAC cases in the RARC group. How about this point?

Since the number of RARC cases is small, we expect another comparison when the number of cases increases.

Author Response

Dear reviewer,

Thank you for your comments. Here I reported here our considerations.

Warm regards,

Filippo Gavi

Reviewer:

1) There is no significant difference, but there may be a difference because there are more NAC cases in the RARC group. How about this point?

Answer: Dear reviewer, thank you for your feedback. You are absolutely right, we suspected it as well and so we decided to put NAC as a variable in the PSM model. However, no significant differences were found.

Round 2

Reviewer 1 Report

Comments and Suggestions for Authors

The paper has been extensively revised and rewritten, and the authors have accepted almost all of the suggestions made. The paper has become clearer and more valuable.